# Are Larger Litters a Concern for Piglet Survival or An Effectively Manageable Trait?

**DOI:** 10.3390/ani10020309

**Published:** 2020-02-17

**Authors:** Sophia A. Ward, Roy N. Kirkwood, Kate J. Plush

**Affiliations:** 1School of Animal and Veterinary Sciences, University of Adelaide, Roseworthy, SA 5371, Australia; roy.kirkwood@adelaide.edu.au; 2Sunpork Group, Murarrie, QLD 4172, Australia; kate.plush@sunporkfarms.com.au

**Keywords:** farrowing, management piglets, pre-weaning, mortality

## Abstract

**Simple Summary:**

In the swine industry, sows are selectively bred for larger litters so, theoretically, more pigs can be sold per year. As producers continue to increase the number of piglets born in a litter, it is necessary to review problems that can arise in larger litters, and whether these issues can be effectively managed and/or require pharmacological intervention. Additionally, this review will reflect on whether selecting sows for larger litter sizes is an ethical concern, regardless of how effectively it can be managed.

**Abstract:**

As sows continue to be selected for greater prolificacy, it is important to review problems that arise in larger litters, and whether these issues can be appropriately managed. Although a proportion of piglets in larger litters can be born underweight, proper supervision around farrowing and adequate colostrum intake has the potential to improve the survival of low-birth-weight piglets and their ongoing growth to weaning. As larger litters can impart greater stress and discomfort on sows, implementing a low-stress environment leading up to parturition may improve sow performance and subsequent survival of piglets. Additionally, treating sows with anti-inflammatory compounds, either dietary or pharmacologically, shows some promise for alleviating sow discomfort and improving piglet survival in larger litters. Understanding that selecting sows for larger litters not only affects piglet survival but the well-being of the sow, the decision to continue selecting for larger litters, regardless of management strategies, remains a topic of ethical concern.

## 1. Introduction

As advances in genetics [1,2,3], reproductive management [4], and nutrition [5,6] continue to increase the number of piglets a sow can produce per litter [7,8], it is important to consider the issues that are associated with larger litters, particularly surrounding piglets and pre-weaning survival. Issues surrounding large litters include the effects of intrauterine crowding and so birth weight variation, piglet hypoxia during delivery and litter-mate competition post-partum. Potential management strategies for improving survival in large litters will be examined, including effective piglet fostering techniques, altering sow environments to reduce stress and the provision of anti-inflammatory compounds, both medicinal and dietary, to alleviate discomfort and improve performance. For this review, litter size is defined as all piglets born in a litter, born dead and alive, that would have contributed to intrauterine crowding during development. In turn, studies with varied interpretations of litter size will also be mentioned. The aim of this review is to identify factors that contribute to high piglet mortality in large litters, and by doing so suggest interventions that reduce the risk of piglet death.

## 2. Issues Surrounding Larger Litters

### 2.1. Intrauterine Crowding and Its Impact on Piglet Development

Although sows have the capacity to conceive larger litters, uterine space and blood supply are limited resources. [9,10,11]. On average, pregnancy is initiated in sows with the presence of approximately 15–20 viable embryos [11]. In an average litter, 9–13 of these embryos will eventually develop into live-born piglets [12] but litters greater than 16 piglets are no longer uncommon in commercial production [13,14]. In larger litters, the uterus of a sow is crowded with embryos. When intrauterine crowding occurs, embryos first to implant can physically restrict the development of later attaching embryos, and this embryonic competition increases with every successful embryonic attachment [15]. Additionally, once the uterus has surpassed normal limits of uterine space, every additional littermate is associated with a reduction in individual fetal growth [16,17]. In turn, larger litters are strongly correlated with a proportion of piglets born underweight (<1.0 kg) [8,10]. Observing the performance of 965 litters, Quiniou et al. [7] found that larger litters had a 33 g decrease in mean birth weight average over ‘normal’ litters at 11 pigs. As low-birth-weight piglets have a larger surface area to volume ratio, they are more susceptible to weakness, hypothermia and hypoglycemia within the first 24 h of life [18]. Thus, low-birth-weight pigs have an increased risk of pre-weaning mortality compared to normal-weight pigs [19,20,21].

Quiniou et al. [7] found that selection for larger litters not only reduced the mean birth weight in the litter, but also the uniformity of birth weight between littermates. Due to embryonic competition, pigs at the beginning of the order are usually heavier than subsequent littermates [19]. Variability within a litter makes it more difficult for low-birth-weight pigs to compete for a teat and ingest an adequate amount of colostrum. In addition, larger pigs compete indirectly with smaller pigs by draining and having more milk directed to their respective teats [22]. This indirect competition between littermates may explain why differences in bodyweight at birth are often maintained and even exacerbated throughout lactation [23]. 

In addition to reduced birth weight, intrauterine crowding can retard the physiological development of the fetus during gestation [24,25]. Intrauterine growth-restricted (IUGR) piglets are not only physically disproportionate at birth with a ‘dolphin-like’ head shape [26] but are also compromised metabolically by immature intestinal development [27,28] and an increased disturbance in inflammatory and metabolic profiles [29]. 

As such, IUGR pigs have a significantly lower capacity for early survival [26,30], and early management should be prioritized to non-IUGR, low-birth-weight pigs [31]. Management strategies for improving survival of low-birth-weight piglets include the provision of an energy source [32], exposure to warmth [33], and assisting with colostrum intake [34]. Low-birth-weight piglets that survive to market weight have been found to have similar carcass quality, meat palatability and prime cut weights as pigs born of normal weight [35]. Prioritizing management towards their survival in larger litters would therefore be worthwhile from not only an ethical standpoint, but an economic one. 

### 2.2. Intrapartum Hypoxia and Farrowing Difficulties

In addition to in utero development, issues can arise for larger litters around farrowing, which can have significant effects on early piglet survival. Problems during the birthing process can lead to an increased incidence of intrapartum hypoxia [36,37]. When a piglet experiences hypoxia during delivery, the concentration of lactate in the blood rises as ATP must be created in the absence of oxygen [38]. Plush et al. [39] found with every piglet born alive in a litter, the concentration of lactate in piglet cord blood increased by 0.18 ± 0.1 mmol. Intrapartum hypoxia is so hazardous to piglets as even temporary deprivation of oxygen can cause permanent damage to the brain and central nervous system [36]. Mota-Rojas et al. [40] found approximately 14% of live-born piglets in commercial production have reduced viability as a result of experiencing temporary hypoxia during delivery. Low viability piglets are less likely to consume colostrum post-partum and have a greater risk of being overlain by the sow [41]. In addition, piglets that experienced near death hypoxia had abnormal respiratory efforts and cardiac rates [42], which negatively affect early viability. Lucia et al. [43] found sows giving birth to more than 12 pigs were twice as likely to have a stillbirth, and eight times more likely to have a dystocia event requiring manual assistance. Issues arise with larger litters as the farrowing process usually takes longer [42,43], which increases the risk of farrowing difficulties [42]. Peltoniemi et al. [44] observed sows with a farrowing duration over 300 min were twice as likely to have a fetal death during or immediately after birth. Therefore, strategies for reducing farrowing duration in prolific sows should be examined. 

Along with increased litter size, farrowing can be prolonged in sows of higher parity [45] as well as for those sows experiencing abnormally high levels of stress around parturition [46]. Abnormally high stress and pain responses during parturition increase circulating catecholamines in sows [47]. As natural inhibitors of oxytocin, higher concentrations of catecholamines can potentially slow or stop myometrial contractions [48] and prolong piglet birth intervals to dangerous levels. Further, although an increase in cortisol is necessary for triggering parturition [49], excessive levels of cortisol may lead to issues during farrowing. As both prolonged farrowing duration [13] and increases in litter size [50] have been found to increase circulating levels of cortisol in the sow, it would be important to review how stress can be minimized leading up to and during parturition. 

### 2.3. Increased Litter Competition and Insufficient Colostrum Intake 

Once born, a piglet’s survival depends on its ability to effectively compete with littermates for a teat to suckle colostrum [34]. The more piglets that there are in a litter, the greater the competition is for teat access, particularly for piglets of lower birth weight and/or viability [22]. Colostrum is the first secretion of the mammary gland, characterized by its richness in dry matter and immunoglobulins [51]. These mammary secretions are essential for extrauterine survival as they provide piglets with a source of heat, digestible energy, immunoglobulins and immune cells [52,53]. As the epitheliochorial placenta of a sow does not permit transfer of antibodies, piglets are reliant on colostrum for maternal passive immunity transfer and protection from infection [54]. Absorption of IgG and immune cells by piglets is dependent on timing of gut closure or visceral maturation and the leakiness of the piglet intestinal mucosa [55]. As well as being vulnerable to pathogens [54], piglets have no brown adipose tissue and only a small amount of energy to allow the shivering reflex [55,56]. The minimum net energy required by a 1.0 kg piglet for heat production is between 900 and 1000 kJ on the first day [54]. Although glycogen body reserves can provide some energy, it only amounts to approximately 420 kJ/kg BW [55] and a colostrum intake lower than 140–150 g is insufficient to meet energy requirements. Devillers et al. [51] observed piglets consuming less than 200 g of colostrum had a pre-weaning mortality rate of 43.4%, whereas piglets who consumed over 200 g had a mortality rate as low as 7.1%. In addition, it was found that piglets ingesting less than 290 g of colostrum had a 15% reduction in body weight at weaning [51], a result supported by Quesnel et al. [56] in their review on colostrum intake and piglet performance. Larger litters do not only have a greater proportion of piglets born underweight, but also a greater variance in birth weights within the litter. Le Dividich et al. [54] found colostrum intake in piglets was reduced by 26 ± 1.6 g for every 100 g reduction in birth weight. As colostrum production is not determined by litter size [42] and the fixed volume of colostrum provided by the sow must be shared amongst all piglets, there is a lesser chance of low-birth-weight piglets ingesting an adequate amount of colostrum [57] and they are likely to be outcompeted for teat access by larger littermates [31].

### 2.4. Increased Incidence of Piglet Crushing by the Sow 

One of the leading causes of early piglet mortality is the crushing or overlay of piglets by the sow [58]. Across pig breeds, litter size is a contributing factor towards higher crushing incidence [59,60] along with increased sow parity [60], sow movement [58], poorer maternal behaviours [60] and reduced piglet vitality [61]. Sows that experience stress and discomfort during the periparturient period are more likely to move around and increase the likelihood of overlay, especially if the sow ‘flops straight down’ from a standing position [58]. As the incidence of sows crushing any piglets was greater for prolific sows, Andersen et al. [62] theorized that the crushing could be a potential strategy to reduce maternal investment in larger litters. As for the piglets, those more susceptible to being overlain are usually weaker and with lower viability [61]. 

## 3. Potential Strategies for Improving Survival in Large Litters

### 3.1. Managing Colostrum Consumption

As low-birth-weight piglets have less energy reserves and a lower capacity for thermoregulation [20,21], they are especially dependent on adequate colostrum intake for survival. Moreira et al. [63] observed the chance of low-birth-weight pigs (800–1200 g) surviving to weaning rose over 89% when they received 200 mL of colostrum (50 mL every 6 h). This finding supports Declerck et al. [34] who found that the correlation between low colostrum intake and reduced pre weaning survival had the greatest effect on piglets in the lower-birth-weight bracket. In addition to its role in early survival, colostrum has a notable effect on the growth and maturation of the neonatal gut [64]. Several bioactive components in colostrum are responsible for activating enzymes along the intestinal brush border and triggering crypt cell proliferation [63,64,65]. Insulin-like growth factor-1 (IGF-1), one of the compounds responsible for gut maturation, is twice as concentrated in sow colostrum as in milk [66], highlighting the importance of colostrum for both early survival and regular development. In turn, closer management around large litters should focus on colostrum ingestion for both low-birth-weight and low viable piglets. Effective strategies include the split suckling technique [67,68] which allows lower-birth-weight pigs the opportunity to suck by temporarily crating larger piglets [68]. When there were more piglets than functioning teats, i.e., teats that provide adequate volumes of colostrum, the pre-weaning mortality rate in the litter was shown to increase from 8% to 14%. In circumstances where piglets must be fostered off a sow, fostering should occur after colostrum ingestion from the maternal sow but before establishment of teat order by littermates [69]. Deen and Bilkei. [70] observed that low-birth-weight piglets had a greater chance of survival in litters when larger piglets were fostered off, and it is recommended that small piglets remain on the maternal sow [68]. 

### 3.2. Inducing Sows for Increased Farrowing Supervision

One of the best management strategies for piglet survival in larger litters is adequate farrowing supervision [71]. If sows give birth during working hours, producers are able to effectively save piglets at risk by keeping neonates warm [72,73], rescuing overlain piglets from under sows [74], encouraging suckling behaviours [72], and assisting sows with farrowing difficulties [73].

To allow for this extra supervision, sows can be induced to farrow using prostaglandin (PG) F2α or analogues (e.g., cloprostenol) [74]. The optimal time to induce farrowing is herd specific, but induction should not be performed prior to two days before the herd average gestation length [75]. The main reason for this is that the saccular phase of lung development only occurs during the last two weeks of gestation [76] and inducing too early will result in liveborn pigs with compromised lung function. To improve the likelihood of sows farrowing 22–32 h post treatment, two PGF2α injections should be administered approximately 6 h apart [75], which increases the proportion of sows farrowing the next working day from 55% to 84% [74,75,76], and thus allows for closer supervision of piglets during and after birth.

### 3.3. Treating Sows with Uterotonics during Farrowing

A uterotonic such as oxytocin is administered to stimulate uterine contractions to shorten farrowing duration [74,76,77]. Although treating sows with oxytocin may reduce farrowing duration [76], it can cause adverse effects for both sow and piglet. When oxytocin is administered before any piglet had been born, farrowing can be prolonged due to the pain of delivery through a potentially incompletely dilated cervix inducing an acute release of adrenaline, potentially inhibiting further uterine contractions [74]. When the cervix has completely dilated, as indicated by the delivery of the first pig, the administration of oxytocin will reduce farrowing duration but can trigger such powerful and long-lasting uterine contractions that it has been linked with greater fetal stress, intrapartum hypoxia and stillbirth [74,76,78,79,80]. Lucia et al. [42] found the use of oxytocin during farrowing increased the incidence of stillbirth and its use was not recommended until a minimum of six piglets had been born. An alternative and less potent uterotonic to oxytocin, carbetocin reduces farrowing duration [81,82,83,84], reduces piglet hypoxia [81,82] and stillbirth rate [83], but was associated with a reduction in piglet circulating protein concentrations [82] and, presumably, colostrum uptake [84]. Rather than providing an alternate uterotonic to oxytocin, other management strategies that may improve farrowing performance should be considered to minimize early mortality in larger litters. 

### 3.4. Reducing Sow Stress to Improve Farrowing Performance

Regardless of the farrowing environment, cortisol concentrations always rise prior to parturition [85]. Although this rise is expected, it is important to evaluate how stress around parturition can be controlled to minimize the risk of farrowing issues. Sows housed in farrowing crates in late gestation have reportedly higher concentrations of plasma cortisol than do sows housed in pens [86] and this may impact farrowing performance [87]. Farrowing crates are commonplace housing for farrowing sows [13] due to their lower space requirements and reduced risk of overlays [88,89]. In comparison to alternative systems like pens, farrowing crates limit the sow’s movement to sitting and standing positions, which can increase stress as sows cannot exhibit natural pre-farrowing behaviours [88,89]. As a way of improving sow well-being and possibly piglet survival, alternative farrowing environments have been investigated, as summarized in Table 1. 

Although sows in open pen systems have an increased tendency to overlay piglets compared to conventional crates [92,93,95], evidence of improved farrowing performance has been observed [50,94,95]. As suggested by Temple Grandin, even if a new farrowing environment could reduce sow stress without compromising piglet survival, the costs and space needed to implement new technologies into commercial production often requires ‘*…more work than doing the research.’* [96]. As such, it is important to review alternative ways to reduce the stress associated with farrowing confinement for sows rearing larger litters in order to improve farrowing performance. 

According to Grandin and Johnson [97], if you cannot give an animal the freedom to act naturally, then you should think about how to satisfy the emotion that motivates the behaviour. If a suitable nesting environment cannot be provided for the sow, there may be another way to satisfy strong nesting desires for confined sows. Baxter et al. found the provision of a comfortable and flexible lying substrate was enough to ‘switch off’ nesting behaviour in sows, including pawing and manipulating surrounding substrate [98]. Damm et al. [99] suggested the sight of a perceived ‘hollow’ in front of a sow’s crate is enough to satisfy nesting behaviour. For crated sows, a change in the surrounding environment may show improvements to sow behaviour, but there is currently little evidence of improvement in early piglet survival [100,101,102,103,104]. 

### 3.5. Provision of Dietary Supplements

During late gestation and lactation, the increased metabolic demand on the sow can elevate the concentration of free radicals and, in turn, the levels of oxidative stress [105]. In addition to negative effects on sow well-being [106], elevated levels of oxidative stress around the periparturient period can impair early lactation output and increase risk of stillbirth [105]. As oxidative stress has been found to increase with litter size [107], it may be a particular concern for hyper prolific sows. The use of supplemental oil with antioxidant properties could be a low-cost strategy for reducing oxidative stress. The supplement’s effectiveness in reducing oxidative stress is affected by oil type, oil quality (i.e., is it oxidized) and dosage. A summary of antioxidant-based oil supplements and their effectiveness on sow performance is presented in Table 2. 

Evidently, oils that stimulated greater release of anti-inflammatory compounds and reduced oxidative stress show a positive effect on sow performance [109,111] and improvements to pre-weaning survival [108,109]. Reducing inflammation can alleviate discomfort in the sow and bring down stress associated with farrowing a large litter of piglets in the confinement of a conventional farrowing crate. 

### 3.6. Provision of Anti-Inflammatory Drugs

Providing an anti-inflammatory drug to peripartum sows would have a more rapid onset of effect than with dietary supplements, and so potentially improve sow and/or piglet performance. Sows experiencing significant discomfort during farrowing show lower circulating concentrations of oxytocin [114] due to elevated cortisol triggering the release of opioids. Lower levels of oxytocin may disrupt rhythmic myometrial contractions [49,115] and thus prolong the delivery of piglets. As well as risk to piglet delivery, pain around parturition may increase the level of agitation and activity displayed by the sow [42]. If anti-inflammatory treatments could diminish this activity, then there would be more opportunities for piglets to suck with a reduced risk of overlay. Anti-inflammatory treatment relieves sow pain by blocking certain stages of the inflammatory process [116]. Treatment can be either non-steroidal (NSAIDs) or steroidal (glucocorticoids), which have different effects on the body due to their different solubility and ability to cross cellular phospholipid bilayers [116,117].

#### 3.6.1. Non-Steroidal Anti-Inflammatory Drugs (NSAIDs)

NSAIDs are weak organic acids that cannot pass freely through the phospholipid bilayer and that inhibit cyclooxygenase enzymes (COX I and II) from converting arachidonic acid into pro-inflammatory prostaglandins [117]. Homedes et al. [118] found the NSAID ketaprofen improved piglet survival at days two to seven postpartum and that this effect was most pronounced in large litters. Other studies involving both ketaprofen and meloxican found no evidence of improved piglet survival [119,120,121,122]. As colostrum and milk production are limiting factors in larger litters, it has been suggested the benefits of ketaprofen were influenced by the sow’s improved milk production [118]. NSAIDs have been found to significantly reduce the incidence of constipation in sows [121], which is important for milk yield due to reduced levels of lipopolysaccharide absorption and increased levels of circulating prolactin [121]. Prolactin is responsible for triggering lactogenesis by stimulating the synthesis of lactose, which is required in high amounts for water to transfer into the mammary alveolar lumen [122]. Further, Mainau et al. [123] found that although piglet survival rates were not affected, treating sows with NSAIDs post-partum improved IgG concentration in the serum of day-old piglets. NSAID may show improvements to lactation output, but it is not certain whether this translates into improved piglet survival to weaning. 

#### 3.6.2. Steroidal Anti-Inflammatory Drugs

Glucocorticoids (GCs) are a class of steroid hormones that assist in reducing inflammation in response to biological stress [124]. The in vivo synthesis and secretion of GCs are regulated by the hypothalamic–pituitary–adrenal axis (HPA) but can also be administered exogenously [125]. As lipid-soluble steroids, GCs diffuse across the cell membrane to stimulate a targeted anti-inflammatory response [126,127]. By passing directly through the cell wall, GCs trigger the ‘switching off’ of genes that code for pro-inflammatory proteins. Inhibiting the production of chemokines, cytokines and the enzymes COX I and II will reduce inflammation and pain in the target tissue [128,129]. As well as delivering a targeted response, GCs relieve inflammation at multiple sites due to their ability to enter any cell type [130], which is why they are a highly effective anti-inflammatory treatment. Duration, dosage, GC type and mode of application all have an influence on the effect on the anti-inflammatory effect [128]. The type of GC is classified by its potency, with plasma half-lives ranging from 80 min (cortisol) to 270 min (dexamethasone) [126]. In addition to the method of treatment, the effect of GCs on a cell is dependent on the rate of absorption as well as the presence of GC receptors. A potent GC like dexamethasone should be the most effective GC for relieving inflammatory pain that arises with farrowing due to its high potency and long half-life.

To exert an effect on genes, GCs must bind to specific receptors present in almost all cells in the mammalian body [125,128] Once activated, GC receptors (GCRs) dissociate from chaperone proteins (e.g., HSP90), translocate to the nucleus and bind to GC-responsive elements for gene expression [126].

In addition to their presence in other organs, GCRs are highly expressed within the placenta [125,128] as they are important signaling hormones during pregnancy [129]. In early gestation, GCs are responsible for detecting available intrauterine space and modifying fetal development. Towards the end of gestation, a natural rise in GCs triggers the maturation of organs to prepare the fetuses for extra uterine life [129,130]. Although GCs can readily cross the placental barrier, the fetus is protected from GC overexposure by placental 11-β-hydroxysteroid dehydrogenase type 2 (11-β-HSD-2) [131]. This enzyme facilitates the conversion of most of the maternal cortisol into inert 11-keto forms to ensure the fetus has a far lower level of circulating GCs than the mother [124,131]. The level of GC exposure a fetus receives during gestation will have lifelong consequences to its physiological ‘programming’ [126,132]. According to Seckl et al. [125], programming refers to the associations between perinatal environmental events and later pathophysiology. Regulating fetal GC exposure is important, as high GC concentrations have been shown to restrict skeletal development, increase the likelihood of IUGR, and impair normal programming responses [109,133]. In humans, treating healthy mothers with dexamethasone during late pregnancy has also shown improvements in the viability of pre-term infants and improves chances of fetal survival [133]. The effects GCs have on the developing fetuses are largely dependent on the timing of treatment, stage of fetal development and level of fetal exposure [123,134]. 

Treating crated sows with glucocorticoids prior to farrowing has potential to not only improve well-being of the sow, but the survival and future growth of piglets. The treatment of sows with glucocorticoids prior to farrowing has the potential to reduce the pain and inflammatory response, thus resulting in improved sow well-being and lactation performance. Due to the ability of glucocorticoids to advance the maturation of visceral organs, glucocorticoids used in conjunction with farrowing induction may also reduce the risk of underdeveloped viscera (e.g., lungs and gastrointestinal system) and improve piglet preweaning growth and survival. However, the appropriate timing and duration of treatment is currently unknown.

## 4. Conclusions

Based on the literature reviewed, it is evident that larger litters provide significant challenges for sow and litter management. The current management paradigm is essentially an economic one, with profit being the primary motivation. The development of hyper prolific sows fits with the economic paradigm as the modern sow weans more pigs available for market. However, perhaps we need a more ethical paradigm and should not be asking how many pigs are weaned, but rather how many pigs died in order to achieve the numbers weaned. The increased risk of low-birth-weight piglets with greater weight variation within the litter makes it very difficult for the smaller piglets to survive to weaning, and many require significant attention and supervision to survive. On many, if not most, sow farms, the necessary supervision is not available. Longer farrowing durations, an increased incidence of stillbirth and intrapartum hypoxia, inadequate colostrum intake and overlain piglets are issues that need to be addressed as the industry evolves to accommodate larger litters. Additionally, understanding that hyperprolific sows have a heightened metabolic demand, greater susceptibility to oxidative stress, usually longer farrowing duration with greater levels of discomfort, and an increased tendency to overlay piglets will all be important for current and future production. Continued research into strategies that will reduce sow stress and allow them opportunities to display natural nesting behaviour and/or movement without the risk of overlay, show signs of improvement in both farrowing and lactation performance. As production science evolves, it is important that the needs of both the sow and piglets are considered. Evidence-based management protocols that show improvements to piglet survival, and that may or may not involve pharmaceutical intervention, should be implemented. 

## Figures and Tables

**Table 1 animals-10-00309-t001:** Alternative farrowing environments and their influence on sow and/or piglet performance.

Farrowing Environments	Observations	Reference
Farrowing crates vs. single open pens vs. group open pens	Higher incidence of crushing in loose housing (both single and grouped) within first three days after birth compared to crated sows	Nicolaisen et al. [90]
Open pens vs. crates	Open penned farrowing increased piglet mortality in three different sow herds	Hales et al. [91]
Open pens vs. crates	Sows housed in open pens had fewer pain related behaviours during farrowing and delivered fewer stillborn piglets	Nowland et al. [50]
Open pens vs. crates vs. crates for 0–4 days postpartum and then moved into open pen.	Crating sows for the first four days postpartum was sufficient to reduce piglet mortality compared to farrowing in open pens	Moustsen et al. [92]
Crates that allow 360 degree movement vs. conventional crates	Piglet survival improved in 360 crates only when a sow farrowed in them for their previous farrowing	King et al. [93]
Freedom farrowing pens vs. crated sows	Freedom farrowing pens allow opportunity to nest build and greater movement. Freedom pens reduced rate of stillbirth and farrowing duration but increased crushing incidence	Gu et al. [94]
‘Schmidt’ pens that provide nesting enrichment vs. conventional crates	Open farrowing pen design showed a tendency towards more crushing, but crated sows tended to have a higher incidence of low-birth-weight piglets	Weber et al. [95]

**Table 2 animals-10-00309-t002:** Factors influencing the effectiveness of oil supplements.

Comparison	Observations	Reference
Oil type (fish oil vs. soybean oil)	Fish oil stimulated greater release of anti-inflammatory compounds and improved rate of pre-weaning survival compared to soybean oil	Yang et al. [108]
Oil type (fish oil vs. olive oil)	Olive oil was more effective in reducing oxidative stress, increased milk fat content and improved rate of pre-weaning survival compared to fish oil	Shen et al. [109]
Oil type (echium oil vs. linseed oil vs. Fish oil)	Fish oil stimulated greater release of anti-inflammatory compounds compared to echium and linseed oils	Tanghe et al. [110]
Supplement (oregano oil) vs. no supplement	Oregano oil effectively reduced oxidative stress on the first day of lactation and increased feed intake of sows three weeks after farrowing	Tan et al. [111]
Oil quality (fresh vs. oxidized)	Fresh corn oil stimulated greater release of anti-inflammatory agents and more effective reduction in oxidative stress compared to oxidized corn oil	Su et al. [112]
Dosage of oil supplement	Increasing the dose of fish and linseed oil from 0.5% to 2% stimulated a greater release of anti-inflammatory EPA into sow serum	Tanghe et al. [113]

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
