# Peer review of "Are Larger Litters a Concern for Piglet Survival or An Effectively Manageable Trait?"

_animals, 2020, doi:10.3390/ani10020309_

Round 1

Reviewer 1 Report

58 Additionally, once the uterus has surpassed normal limits of uterine space, every additional littermate is negatively correlated with individual fetal growth [20]. What does "negatively correlated" mean in this context?

84. Therefore, it is worth prioritizing management of low birthweight pigs, particularly ensuring adequate colostrum intake. Why is colostrum most important?

224: Hyper-prolific sows have very high metabolic demands during gestation and lactation [102, 103],

Reference 102 and 103 are about loose housing and not explaining the dimension of "very high". Check if reference numbers has been accidentally changed?

3.3. Treating sows with uterotonics during farrowing

More structure needed. Describing the effekt of exogenous oxytocine before farrowing followed by exogenous oxytocin during farrowing is suggested.

220. Although changing farrowing the environment may change sow behaviour, the literature points to little improvement in early piglet survival [98-101]. Some words are missing in this sentence. 

230 Supplementing the diet with olive oil was also found to reduce overall movements by sows after farrowing, which may be due to its anti-inflammatory properties [107].

Ref. 107 is about oregano oil (not olive). And the effect was measured as oxidative stress.  Check the adjacent references too.

232 Sows experience inflammation during farrowing [108], as well as through changes in physiological state from gestation to lactation [109]. As inflammation has the potential to negatively impact 233 lactation performance, sow behavior and overall piglet survival, anti-inflammatory treatments are 234 worthy of investigation. 235

this text belong under the headline 236 and not above. 

236

3.6. Provision of anti-inflammatory

3.6. Provision of anti-inflammatory drugs

This chapter needs revision, for the understanding if pain from something has an effect on farrowing, or if farrowing causes pain leading to disrupted farrowing.

254 Homedes et al [112] found the NSAID Ketaprofen improved piglet survival at day two to seven postpartum, however, other studies involving both Ketaprofen 255 and Meloxican have found no evidence of improved piglet survival [108,113-115]. Homedes et al. 256 [112] found the differences in piglet mortality between treated and non-treated sows was greater as 257 litter size increased. 258

Suggestion: Homedes et al [112] found the NSAID Ketaprofen improved piglet survival at day two to seven postpartum,and that this effect was most pronounced in larger litters. Other studies involving both Ketaprofen 255 and Meloxican  found no evidence of improved piglet survival [108,113-115].

266. NSAID may show improvements to lactation output, but there has not been sufficient evidence of the treatment??? the number of piglets weaned per litter.

Some words are missing here.

Reference 118 describes the effect of a NSAID, but the reference 118 is used in the chapter on Steroids.  

301. Treating healthy mothers with dexamethasone during late pregnancy has also shown improvements in the 302 viability of pre-term infants and improves chances of fetal survival [130, 131].

For a clear understanding start the text with "In humans" 

76 Vande Pol, K. D.; Cooper, N.; Tolosa, A.; Ellis, M.; Shull, C. M.; Brown, K.; Alencar, S. PSV- 4 Effect of drying and/or warming piglets at birth on rectal temperature over the first 24 530 hours after birth. J. Anim. Sci. 2019, 97, 157-158.

Check if this reference describe "pulling overlain piglets from under sows [40, 76]" Reference 40 is OK in this aspect.

Line 224 and line 230: The reference numbers to the text in this section do not fit in with the right papers in the reference list. The author should double check all references.

Author Response

Thankyou very much for reviewing my paper. I really appreciate how much attention you put into the critiques. As a young researcher it is really helpful for me to receive critical feedback. 

Please see attached my updated review. I hope it is now acceptable for publication. 

Kind regards,

Sophia. 

Reviewer 2 Report

In the present review, the authors deal with large litters in piglets, related welfare problems and ways to meet these challenges. From a formal point of view, the manuscript is well written and references were collected diligently and broadly. However, concerning the content, despite what the title or the submission to the journal ANIMALS leads one to expect, the author do not elaborate on welfare or behavior (crushing?) in detail, but emphasize medical treatment with glucocorticoids and NSAIDs to improve sow’s and piglets’ performance.

If this is a sustainable way to increase animal welfare can -and should- be discussed and the attitude might differ from country to country, but some other, and more critical, aspects concerning welfare and large litters are missing, for instance a discussion on breeding strategies and sow characteristics. In the current version, the breeding goal of large litters is accepted rather uncritically. One can doubt if the use of medication to handle the problems associated with large litters is a sustainable and acceptable way to “solve welfare problems”.

Before this paper can be considered as suitable for ANIMALS, these points should be discussed. The current point of view is focusing mainly on veterinary, curative aspects. A submission of this work to a “pure” veterinary journal, probably species-specific for pigs, might be considered. Some further suggestions for improvement:

Change of title, “Issues” is very general, the hint to medical treatments is recommended Abstract: Use of abbreviations (GCs) should be avoided In the abstract, you mention “alternate farrowing environments” in a context that is not very clear: The alternatives aim at improving sow’s welfare (no restrictions etc); the increased piglet mortality due to crushing is a problem in those systems that cannot be solved by medication. Line 40: non-infectious: infections can also play a role. Even if they cannot be discussed in detail in this paper, a general good hygiene and low infection pressure can help to increase piglet survival. You mention management of colostrum intake, but what about milk intake in general? What about freezing colostrum of other sows? Artificial fostering systems? Critical discussion on use of medical treatment?

Author Response

(The authors gave the same response as above.)

Round 2

Reviewer 2 Report

In this revised version, the authors improved the manuscript significantly. They put a lot of effort in improving the text according to the reviewers’ suggestions, and answered the raised questions adequately. In my opinion, the manuscript can be published in its present from.

Author Response

Although larger litters are a real problem from a commercial aspect of swine production with welfare concerns intertwined, I genuinely have concerns about publishing this in ANIMALS as written. There is science out there indicating that larger litters are associated with welfare concerns, but at times the scientific details are missing and result in a bias conclusion.  The review at times makes broad conclusions without providing specifics that support their claim that larger litters = ethical concerns.  

The reviewers accepted the paper. If that is the final decision, the manuscript will need to be edited, and based on my assessment; the authors did not sufficiently address the concerns of one of the reviewers.  Simply justifying significant concerns with the authors' opinions/and or constraints of the journal (e.g., would make it too lengthy) is not sufficient.

We apologise for the wording which we agree needs improvement. The reviewer asks about breeding management, which is something we totally agree with. Anecdotally, I am informed that the Dutch are rolling back their litter size targets in order to reduce an unacceptable preweaning mortality. However, I am not aware of the original source and so am unable to reference the point. As for increasing emphasis on lactation management, we refer the reader to the Alexopoulos et al review, which is cited as reference 68.

One of the reasons infections are increased in larger litters is due to inadequate colostrum uptake by piglets of lower vitality and birthweight. Colostrum provides piglets with immunity, so inadequate intake leaves these piglets to a greater susceptibility of infection.

Line 111: Once born, a piglet’s survival depends on its ability to effectively compete with littermates for a teat to suckle colostrum [34]. The more piglets there are in a litter, the greater the competition is for teat access, particularly for piglets of lower birthweight and/or viability [22]

Line 116: As the epitheliochorial placenta of a sow does not permit transfer of antibodies, piglets are reliant on colostrum for maternal passive immunity transfer and protection from infection [54].

One of the reviewers raised valid concerns, and the authors' response in my assessment did not successfully address concerns raised by a reviewer. The title change made so that it is more appealing to the reader does not adequately address this concern.  The title should be reflective of the "story" being told in the review--I think it is more confusing and implies bias.   

The title is now: Are larger litters a concern for piglet survival or an effectively manageable trait?

 I do believe this is a title that is reflective of the paper – first part reviews the issues that can arise from larger litters and the second part reviews how different studies have gone about trying to reduce the issues raised.

Regardless, this is an important area, but the conclusions the authors are making without providing essential details from the paper they are referencing is concerning.  The way the authors utilized review papers within this manuscript is also somewhat unconventional.  For example (L91), Alonso-Spilsbury is referenced, but Moto et al. paper should be referenced instead.

Corrected.

Another example (L98-99), one of the citations, focuses on the effects of larger litter sizes on the immunity of the sow and her piglets, and yet it is used to support the topic at hand.

With this point, where it is mentioned that prolonged farrowing duration can increase neonatal mortality was not supposed to be misleading:

Issues arise with larger litters as the farrowing process usually takes longer [42, 43], which increases the risk of farrowing difficulties [42]. Peltoniemi et al. [44] observed sows with a farrowing duration over 300 minutes were twice as likely to have a fetal death during or immediately after birth. Therefore, strategies for reducing farrowing duration in prolific sows should be examined.

This is not supposed to sway the reader to a certain point of view but was rather trying to create a connection that larger litter size can cause an increase in farrowing duration which can create greater issues with the farrowing process as shown by Peltoniemi et al. in their study focusing on farrowing duration and piglet stillbirths.

 It is important to use these papers regardless of the year published, but the authors need to be specific in some areas of their review article.  For example, one citation used to support the relationship between increased incidence of hypoxia and larger litter sizes. In that particular paper, the avg litter size was 9, and the focus was on hypoxia but implied it was due to larger litter size.

They are leaving details out especially when older papers are used to support their theory results in the wrong information.  Litter size of 9 is different than litter size of 12 or 16, etc. and when you leave this information out (or do not define litter size) then one concludes incorrectly.  

Some papers used are not focused on litter size at all, but are important to explain how intrapartum hypoxia can affect survivability and vitality of piglets. When a point is made that litter size can affect the incidence of hypoxia, the number of pigs is specified (over 12 in the study by Lucia et al.)
It is not in my interest to be biased and again, I do not want to be misleading to the reader. This is why I have also mentioned that hypoxia can be prolonged in sows of higher parity as well as sows experiencing high levels of stress.

The paragraph is not trying to say hypoxia is directly correlated to an increase in litter size. The paragraph reviews what hypoxia is, what hypoxia does to a piglet and how an increase in litter size is one contributing factor.

Other times the wrong papers are cited under the wrong subheadings, or the information provided is inaccurate.  

This has been corrected.  

Another example 142-144. The authors use these references (66) to imply that larger litter size results in increased crushing.  Crushing was a problem before the increase in litter size.  In the Weary citation (in this section), they focused on pen design per se.  They concluded multiple factors contribute to crushing and sow movement including pen design, sow parity, and so forth.  

Line 138 has been edited to show that litter size is one contributing factor, but not the only factor that effects incidence of crushing: Across pig breeds, litter size is a contributing factor towards higher crushing incidence [59-60] along with increased sow parity [60], sow movement [58], poorer maternal behaviors [60] and reduced piglet vitality [61].

The authors believe larger litter = ethical concerns and that medical intervention is the best solution.  If so, scientific data should provide support for this.    

I don’t believe the paper is inferring that medical intervention is necessarily the best solution, just that there is a potential there. Concluding that changes in a sows’ surrounding environment has made little improvement to sow behaviour is not swaying the reader into giving up on this idea, but rather pointing out that current designs do not show an improvement to piglet survival.

Again, the idea that there may be easier ways to reduce stress over completely refurbishing a piggery is not a point that is supposed to dissuade the reader, it was just there to open up discussion on whether other strategies could be researched, as a way to lead into the next points (anti-inflammatory treatments). 

As written, the take-home message is that larger litters are the main cause of all the other problems discussed in the paper, which is not true. Their potential strategies for improving survivability are not practical and some are not validated with science.   For example, giving a glucocorticoid injection is unreasonable from a practical standpoint (it is a good scientific tool) but not a good management tool --- frequently, when we solve one welfare issue, we create another.  It is not one factor that contributes to the welfare concerns associated with larger litters, but many including system, genetics, etc.,

Line 337: “… Continued research into strategies that will reduce sow stress and allow them opportunities to display natural nesting behaviour and/or movement without the risk of overlay, show signs of improvement in both farrowing and lactation performance. As production science evolves, it is important that the needs of both the sow and piglets are considered. Evidence-based management protocols that show improvements to piglet survival, and that may or may not involve pharmaceutical intervention, should be implemented…”

In the conclusion, we believe no idea is necessarily driven into the reader, and the authors touch on pharmaceutical intervention as a possible idea ‘…may or may not be implemented..’ as a way to open up personal thought by the reader.

This manuscript is a resubmission of an earlier submission. The following is a list of the peer review reports and author responses from that submission.

Round 1

Reviewer 1 Report

Conflict of interests: I am working in the area of exploiting prolific breeds, and thus see a number of environmental and economical benefits of high litter size.

Reviewers overall evaluation of this text: there is a lack of correspondence between the title of the paper, the abstract, he structure of the text and the conclusion. 

Reviewers additional comments: Lack of definitions (eg. of prenatal survival) and a lack of numbers, that could help the reader to understand the general level of prenatal survival, and the reductions in prenatal survival caused by the different causes for prenatal survival, which are discussed in the text.

The headline covers preweaning piglet survival and prenatal glucocorticoids

The abstract concludes, that effectively managing low birth weight piglets and implementing strategies to improve sow performance around farrowing are the most relevant factors to improve pre-weaning survival. Neither the simple summary nor the abstract mentions glucocorticoids.

The conclusion is, that treatment of crated sows with glucocorticoids has the potential of improving the well being of the sow and the survival of the piglets.

The reviewer postulates, that a major problem in the text is the lack of numbers. There is no indication of the individual relevance of the described factors reducing prenatal survival. What is normal and high litter size? What is the amount of piglets dying due to the different factors described in the text. Thus the reader has no chance to select the most relevant factors to focus on. It is accepted by the reviewer, that overlaying of piglets, sows not producing milk and infections are not described as potential factors reducing piglet survival, but these factors should be mentioned, with a note why they are not relevant for this paper.

There is a lack of definitions in the paper: Are still born piglets included in prenatal mortality in this paper? What is the size of small, normal and large litters? What is normal birth weight?

Examples on insufficient text:

Line 33. Along with varied birth weights, selection to increase litter size has been accompanied by an increase in piglet preweaning mortalities, particularly in the first three days postpartum [4-6]. Comment: Did these authors all specify, that it was particularly in the three first days, that piglet mortality was increased in large litters?

Line 74. Farrowing duration can be influenced by parity, gestation length, increased stress, and pain associated with parturition [25]. Reviewer:  Is farrowing shortened or prolonged, when it is influenced?

Line 75. Fear, anxiety and pain during parturition will increase circulating catecholamines, which have the potential to slow or stop myometrial contractions and prolong piglet birth intervals to dangerous levels. Reviewer: This is how to do it. The physiological effect of fear is described, and the final effect on the piglets is clear. It would be optimal with information on the lenght of "dangerous birth intervals". The references are missing.

 Line 81 In general, primiparous sows are more sensitive to environmental factors and show poorer reproductive performance(Rewiever: meaning higher piglet mortality ??) than multiparous sows thus their adequate supervision during farrowing would be important (26). Reviewer: One reference here is not enough. Some investigations indicate, that farrowing in primiparous sows is quicker than in older sows, and that the percentage of still born piglets is lower than in older sows, thus suggesting supervision for older sows or sows which previously had problems farrowing/many still born piglets.

Line 117 . As well as being vulnerable to pathogens, piglets have no brown adipose tissue and only a small amount of energy to allow the shivering reflex.  Reference is missing.

283: 3.2 Non-steroidal anti-inflammatory drugs (NSAIDs). Comments: Effects of NSAID´s on piglet preweaning survival is not directly mentioned in the section. The conclusion of the section only covers lactation performance and constipation.

Line 294. NSAID use may have a significant effect on constipation, but the effectiveness on lactation performance is inconsistent. Reviewer: References are missing to the papers describing NSAID´s effect on lactation, eg. measured as piglet growth in sows treated with NSAID's

321. In early gestation, GCs are responsible for detecting available intrauterine space and modifying fetal development. References are missing. 

Cortisol has its own section beginning in line 211. Why not included in the section on other glucocorticoids?

4.3. The role of synthetic glucocorticoids in early neonatal survival is missing a reference to Cassar et al. (ref. 33),who tested a GC treatment in a controlled study? There must be other references to studies performed in practice, which should be included here. 

In the fetal pig, GCs have been shown to affect the villus-crypt architecture along the intestines and to induce digestive enzymes along all parts of the intestinal tract [78]. Comment: The text does not indicate, which positive and negative effects these two factors (affect crypt architechture and induction of enzymes) will have for survival

Ref. 8. You forgot the authors of this chapter: Nissen, P. M.; Oksbjerg, N.  The following names are editors, and should be mentioned after the title of the book: Editors: Greenwood, P. L.; Bell, A. W.; Vercoe, P. E.; Viljoen, G. J. Quantification of Prenatal Effects of Productivity in Pigs. IN Managing the prenatal environment to enhance livestock productivity 2010, 1, 37-69. DOI: 436 https://doi.org/10.1007/978-90-nd. This book is used as the only reference about number of piglets born in a litter. In the book they write, "Pigs are a litter bearing species that give birth to an average between 9 and 13 pigs per litter depending on breed and country". This seems to be changed in the text in Line 42 into "Around 10-12 of these embryos will eventually develop into live born piglets (8)".  Use of references, where ovulation rate and litter size are corresponding, will give a better understanding of embryo mortality and litter size dynamics, if the authors still find embryo mortality relevant to this text about prenatal survival.

Authors suggestion. The paper is partly a review of causes of EARLY piglet prenatal survival and partly a review of the  physiological effect on new born piglets, when exposing the piglets to natural or synthetic glucocorticoids. A more strict text focusing on only one of these factors, will be much more clear to the reader. Eg. "Early postnatal piglet survival under the effect of external glucocorticoids". Describe the variables concerning piglet mortality, which may be due to immature piglets. Cover the frequency, symptoms and physiology behind the factors. Maybe also pain in the sow. Evaluate the possibility to improve the maturity in those piglets that has the risk of being born immature. Finish with the result of relevant trials mentioned in literature. End up with your suggestion for timing and dose for a better treatment. Be specific concerning numbers.

Reviewer 2 Report

This is an interesting review that brings together a diverse set of ideas around the common theme of increased pre-weaning mortality due to a low birth weight phenotype.  The link between increased cortisol tone driven by the stress created by using farrowing crates, and  negative outcomes for parturition, lactogenesis, milk let-down  and ultimately piglet survival, are described in detail. A diagrammatic "cartoon" describing these interactions would be very helpful to the reader.

The authors conclude with suggestions about the potential benefits of iniversal exogenous GC treatment to ameliorate the problems identified. They are encouraged to indicate whether this is their preferred alternative, compared to the use of improved farrowing environments. In the context of the IUGR pig within a more normal litter, some further thoughts about the timing of their proposed treatment pre-natally would also be valuable.  

In the context of the presented review, the authors specifically focus on the low birth weight (IUGR) pig that may be born in a large litter of otherwise more normally developed littermates. Having read the review several times, it appears that many of the concepts discussed, and proposed remedies, would equally apply to entire low birth weight litters, born to sows with a repeatable low birth weight phenotype. These light weight litters also make a considerable contribution to the increased pre-weaning mortality reported by the industry.  At least a brief reference at the start of the review to this alternative source of low birth weight pigs would be useful.

My other specific comments are presented in the attached file which links text highlights to specific comments and questions.

Reviewer 3 Report

I considered that the review "Pre-weaning piglet survival and the influence of prenatal glucocorticoids" shows serious deficiencies that do not make possible its publication in this journal.

Although it is a review, most of the articles cited are prior to 2014. There are no references, except 4, from the last 6 years. If the reason is the non-existence of publications in this period, the authors should have indicated and justified it. This would remove interest in the subject.´

The introduction, although concise and clear, does not the reader towards the purpose of the paper. And the objective of the paper does not indicate the influence of prenatal glucocorticoids. However, the conclusions are focused on glucocorticoids

The paper is not well structured. The main causes of survival (or mortality) in the sows should have been indicated and they should have been described after. For instance, aspects such as the management of the sow during lactation, crushing of piglet are the first cause of mortality, it is not addressed.

The sections are not balanced. For instance, point 2.4 is very extensive and repetitive, while points 2.2 and 2.3 are very concise and confuse

Section 3 is entitled "factors affecting lactation performance" however section 3.1 is a description of the main lactogenic hormones, without mentioning at any time of their relationship on piglet survival.

Why was a special section written for “Reasons for increased cortisol concentrations”? What is the relationship with piglet survival?.

Again, what is the relationship between 3.2.1 and “lactation performance” and piglet survival?

The lack of structure of the paper reappears with the presence of section 3 (should be point 4) “the use of anti-inflammatory agents”. Why this section? What is your justification? What is the relationship with the objective of the paper?

Section 4 (should be section 5): The role of glucocorticoids in early piglet survival.

Why early? Not pre-weaning?

The authors would have to justify because they focus on glucocorticoids.

The conclusions do not match the stated objectives. The authors over conclude (lines 390-392)

Moreover, the inclusion of tables, figures, schemes could improve the paper.